# Costs, benefits, and cost-benefit of Collaborative Assessment and Management of Suicidality versus enhanced treatment as usual

**Phoebe K. McCutchan**[1]*, **Brian T. Yates**[1], **David A. Jobes**[2], **Amanda H. Kerbrat**[3], **Katherine Anne Comtois**[3]

**1** Department of Psychology, American University, Washington, DC, United States of America,
**2** Department of Psychology, The Catholic University of America, Washington, DC, United States of America,
**3** Center for Suicide Prevention and Recovery, Department of Psychiatry and Behavioral Sciences, University of Washington, Seattle, WA, United States of America

* phoebe.mccutchan@gmail.com

**Data Availability Statement:** Public sharing of data used in this study is prohibited under the protocol approved by the U.S. Army Medical Research and Development Command Office of Research

## Abstract

Suicide rates have been steadily increasing in both the U.S. general population and military, with significant psychological and economic consequences. The purpose of the current study was to examine the economic costs and cost-benefit of the suicide-focused Collaborative Assessment and Management of Suicidality (CAMS) intervention versus enhanced treatment as usual (ETAU) in an active duty military sample using data from a recent randomized controlled trial of CAMS versus ETAU. The full intent-to-treat sample included 148 participants (mean age 26.8 years ± 5.9 SD years, 80% male, 53% White). Using a micro-costing approach, the cost of each condition was calculated at the individual level from a healthcare system perspective. Benefits were estimated at the individual level as cost savings in past-year healthcare expenditures based on direct care reimbursement rates. Cost-benefit was examined in the form of cost-benefit ratios and net benefit. Total costs, benefits, cost-benefit ratios, and net benefit were calculated and analyzed using general linear mixed modeling on multiply imputed datasets. Results indicated that treatment costs did not differ significantly between conditions; however, CAMS was found to produce significantly greater benefit in the form of decreased healthcare expenditures at 6-month follow-up. CAMS also demonstrated significantly greater cost-benefit ratios (i.e., benefit per dollar spent on treatment) and net-benefit (i.e., total benefit less the cost of treatment) at 12-month follow-up. The current study suggests that beyond its clinical effectiveness, CAMS may also convey potential economic advantages over usual care for the treatment of suicidal active duty service members. Our findings demonstrate cost savings in the form of reduced healthcare expenditures, which theoretically represent resources that can be reallocated toward other healthcare system needs, and thus lend support toward the overall value of CAMS.

Protections as well as Institutional Review Boards at Dwight D. Eisenhower Army Medical Center, The Catholic University of America, and the University of Washington. Additional restrictions apply to the availability of these data in order to protect participants' privacy. Requests for access from interested researchers may nevertheless be considered, subject to the terms and conditions of the request and in compliance with the applicable regulations. Requests may be directed to the Office of Sponsored Programs and Research Services, The Catholic University of America, 213 McMahon Hall, 620 Michigan Ave., NE, Washington, DC 20064; Phone: 202-319-5218; Fax: 202-319-4495; Email: CUA-OSP@cua.edu.

**Funding:** The current study represents an unfunded secondary analysis of a randomized-controlled trial; the primary trial was supported by the Department of the Army through federal grant W81XWH-11-1-0164, awarded and administered by the Military Operational Medicine Research Program (MOMRP). The authors received no specific funding for the present work.

**Competing interests:** The authors have read the journal's policy and have the following competing interests to declare: DAJ receives book royalties from the American Psychological Association Press and Guilford Press, and is the founder of CAMS-care, LLC (a professional training and consultation company; https://cams-care.com/). There are no patents, products in development or marketed products associated with this research to declare. This does not alter our adherence to PLOS ONE policies on sharing data and materials. Authors PM, BTY, AHK, and KAC have no competing interests to declare.

## Introduction

Amid unprecedented increases in active duty military suicide rates in recent years, the U.S. Department of Defense (DoD) has recognized suicide prevention to be a top military priority [1,2]. Active duty service members have demonstrated continuously rising suicide rates across all branches of service. In 2019, the suicide mortality rate for the Active Component (i.e., full-time service members) across all services was 25.9 per 100,000, representing a per-year rate ratio of 1.04 from calendar years 2011 through 2019 [3]. These trends are particularly alarming in that they represent the first time in recorded history that the U.S. military suicide rate has equaled or exceeded that of comparable U.S. general population cohorts, beginning in 2008 and continuing to date. Indeed, the 2019 suicide mortality rate for the Active Component was statistically indistinguishable from the 2018 U.S. population rate after adjusting for age and gender [4]. Although the exact causes of the increase in military suicides are uncertain, the most consistently reported factor appears to be increasing co-occurrence of mental health conditions such as posttraumatic stress disorder (PTSD), depression, and substance use [5–9].

In response to this alarming increase in military suicide rates, the DoD has collaborated with other government and private agencies to make considerable investments in research geared toward determining the risk factors and correlates of suicidality and developing effective prevention and management strategies [10]. New interventions aimed specifically at managing suicidality are emerging, but additional research is needed to carefully evaluate the effectiveness of these interventions in rigorous, well-powered studies in order to inform clinical practice guidelines and facilitate integration into routine clinical care.

One promising suicide-focused approach is the Collaborative Assessment and Management of Suicidality (CAMS) [11–13], a therapeutic framework that aims to identify and address patient-articulated "suicidal drivers." To date, CAMS has amassed a substantial evidence base demonstrating its effectiveness in reducing suicidal ideation, overall symptom distress, depression, and hopelessness, including several correlational and quasi-experimental studies [14–21] and five randomized controlled trials (RCTs) across a variety of settings [22–26]. A recent systematic review of CAMS reported it to be a promising approach to managing suicide risk and deliberate self-harm in adults [27], although it noted a high degree of heterogeneity and attrition across existing studies and emphasized the need for additional high-quality studies, particularly RCTs and meta-analyses. Further, a more recent meta-analysis reported that CAMS treatment resulted in significantly lower suicidal ideation and general distress compared to alternative interventions, concluding that it is a well-supported intervention for suicidal ideation according to Center of Disease Control and Prevention criteria [28].

In addition to its clinical effectiveness, another important factor requiring further examination is the cost of the CAMS intervention, particularly in comparison to alternatives. As with any health intervention, understanding the costs of delivering CAMS in relation to both clinical and monetary outcomes produced can be seen as an essential component of informed decision-making by policymakers, healthcare administrators, and even clinicians. The economic reality, especially in a population-based system such as the Military Health System (MHS), is that resources allocated to one intervention are then no longer available for other needed services [29,30]. Therefore, it could be helpful to examine the *value* of an intervention (i.e., costs relative to outcomes) compared to that of other possible alternatives in order to "provide the best to the most for the least" [31].

Preliminary analyses using archival data in one retrospective non-randomized controlled comparison study of CAMS vs. treatment as usual at two U.S. Air Force outpatient clinics found that costs of the treatment conditions did not differ [16]. However, CAMS patients utilized significantly fewer medical services (e.g., emergency room visits, medical appointments,

and minutes spent in these settings) after initiating suicide-related mental health treatment, indicating relative cost savings in the form of reduced healthcare expenditures. Extrapolating the observed utilization rates to the anticipated number of suicidal patients presenting to the clinics over the next fiscal year ($n = 75$), the authors estimated a potential cost savings of approximately $32,500 ($434.25 per patient on average; 2004 dollars). Extrapolating across the total 80 Air Force clinics worldwide, cost savings estimates approached $2 million per year [16]. Still, the significant methodological limitations of the study must be noted, including a small and unequally distributed sample size (N = 55; n = 25 for CAMS and n = 30 for treatment as usual), lack of random assignment, and lack of fidelity measures. Consequently, it is premature to definitively conclude based on these analyses that CAMS is associated with economic advantages compared to treatment alternatives.

Although these preliminary findings were encouraging, additional examinations using more robust study designs, detailed costing methods, and systematic evaluation of treatment benefits are warranted. The purpose of the current study was to extend previous analyses by examining the costs, benefits, and cost-benefit of CAMS compared to enhanced treatment as usual (ETAU) for the treatment of suicidality in active duty Soldiers using data from a recent clinical effectiveness RCT. Results of the primary RCT indicated that CAMS treatment was associated with significant postbaseline improvements in suicidal ideation and suicide attempt behaviors across one year of follow-up. However, the ETAU condition evidenced comparable improvements, with the exception of a significantly greater reduction in probability of suicidal ideation at 3-month follow-up observed in the CAMS condition [24]. Examination of treatment costs and benefits become especially useful in such scenarios where two treatment alternatives appear similarly effective, to select the intervention which maximizes return on investment.

In the current secondary cost analysis, treatment costs (i.e., the value of resources required to implement the intervention) were determined using micro-costing and gross-costing techniques. Benefits of the intervention (i.e., the value of resources generated or saved as a result of the intervention) were assessed as potential cost savings based on healthcare expenditures across multiple categories of services. Finally, cost-benefit (i.e., the relationship between the value of resources used and the value of resources saved by an intervention) was examined using cost-benefit ratio and net benefit metrics [32].

Previous analyses found no significant differences in direct costs (i.e., resource use directly attributable to the intervention) for CAMS compared to usual care [16]. Although CAMS as delivered in the current study required more resources to support the training and consultation activities performed over and above that of the usual care condition, the costs associated with these activities were anticipated to be relatively minimal at the individual level; therefore, it was hypothesized that CAMS would be no more costly than ETAU at each follow-up timepoint (Hypothesis 1).

With regard to benefits, evidence suggests that suicidality is associated with increased health service utilization in active duty Soldiers [33–35]. Because CAMS was anticipated to lead to greater and more rapid clinical improvement in suicidality than ETAU, we predicted that health service utilization would decrease correspondingly. Indeed, usual-care patients have been found to attend significantly more medical appointments compared to CAMS patients over a one-year period [16]. Consequently, it was hypothesized that CAMS would evidence greater benefit (i.e., cost savings) in the form of lower between-group cumulative healthcare expenditures at each follow-up timepoint (Hypothesis 2) and greater reduction in within-subjects past-year healthcare expenditures at 12-month follow-up (Hypothesis 3). Because CAMS was predicted to be less costly and more beneficial than ETAU, we further anticipated that CAMS would be associated with greater cost-benefit than ETAU, as evidenced by significantly

greater cost-benefit ratios (Hypothesis 4) and significantly greater net benefit (Hypothesis 5) at 12-month follow-up.

## Materials and methods

### Trial design

The current study was a secondary data analysis of the DoD-funded "Operation Worth Living" (OWL) study, an RCT of CAMS versus ETAU for suicidal Soldiers. The trial was conducted in the Department of Behavioral Health at an Army Medical Center on an infantry military installation in the Southern United States. Participants were randomly assigned to either CAMS or ETAU matched on histories of suicide attempts, medication class, severity of physical injury or disability, and current enrollment in outpatient behavioral health treatment (i.e., psychiatric, clinical psychology, or social work services). Participants were assessed on clinical variables and service utilization measures at baseline and 1-month, 3-month, 6-month, and 12-month timepoints after baseline. Details about the trial design, including study procedures and power analysis, are described elsewhere [24]. All study procedures were approved by The Catholic University of America, University of Washington, and Eisenhower Army Medical Center Institutional Review Boards. The current archival study was approved by the American University Institutional Review Board (IRB-2017-75).

### Participants

Soldier participants in the study were 148 active duty U.S. infantry Soldiers with current suicidal ideation recruited via provider referral from the community behavioral health clinic, emergency department, and inpatient psychiatric unit. Soldiers were eligible to participate if they were over 18 years old, English-speaking, and had significant suicidal ideation (i.e., a score of $\geq$ 13 on the Scale for Suicidal Ideation-Current) [36]. Soldiers were excluded if they were a member of the Warriors in Transition unit (a unit providing support to soldiers being treated for chronic and/or severe injuries who cannot yet return to work), pregnant, exhibiting significant psychosis or cognitive or physical impairment, judicially ordered to treatment, or ineligible for behavioral health care at the military installation.

On-site clinicians were also consented participants in the study as they completed routine assessments. Eligible clinicians were assigned to either the CAMS or ETAU condition based on their relative self-reported allegiance to treatment models for managing suicidal patients; clinicians reporting weaker preference for a specific treatment were assigned to the CAMS condition to ensure high adherence of ETAU clinicians to their existing approach. The final sample of study therapists participating in the intent-to-treat phase of the trial ($n$ = 4 per treatment condition) consisted of seven licensed clinical social workers and one masters-level mental health counselor. Mean years of practice experience since professional degree were 5.0 ($SD$ = 4.7) for ETAU therapists and 13.25 ($SD$ = 7.4) for CAMS therapists.

### Treatments

CAMS is best described as a therapeutic framework that accommodates a wide range of theoretical orientations and techniques. A core tenet of CAMS therapy is an empathetic and collaborative therapeutic relationship geared toward developing a shared understanding of the patient's suicidality. Guided by the Suicide Status Form (SSF), the fundamental goal is to engage the patient in identifying and addressing the causes ("drivers") that compel him or her to consider suicide using appropriate driver-focused interventions, subsequently reducing suicidal ideation and behaviors as coping mechanisms are increased [11,13]. In the current study,

the CAMS intervention consisted of approximately 4 to 11 weekly individual sessions (after the initial session, CAMS concludes after three consecutive sessions with resolved suicidality) following the suicide-specific CAMS framework [13].

In ETAU, clinicians provided typical care in as many sessions as appropriate to their treatment approach and theoretical orientation without constraint by the study, reflecting the real-world conditions examined within an effectiveness framework to maximize generalizability. This treatment condition was considered "enhanced" treatment as usual in that it specified a minimum number of four treatment sessions, providers had access to consultation and supervision as needed, and Soldiers were routinely engaged and evaluated at the study assessment intervals [24].

## Measures

The Scale for Suicide Ideation-Current (SSI-C) [36], a 19-item interviewer-rated measure assessing suicidal ideation at its highest intensity over the past two weeks, was used to assess suicidal ideation severity (total score of 0–38) and resolution (i.e., a total score of zero). The SSI-C has demonstrated good convergent and criterion validity in previous research in psychiatric patients [36] and evidenced high internal consistency in the current study ($\alpha$ = .88).

Medical and behavioral health service utilization was assessed using the Treatment History Interview-Military version (THI-M), adapted from the Treatment History Interview (THI) [37] for use in the military healthcare system. The THI-M is an interviewer-administered measure that captures the history (e.g., service frequency, duration, and provider type) of outpatient psychotherapy and counseling visits, crisis medical services, medical provider visits, and medication use. The THI has been found to have high convergent validity with medical records ($r$ = .99). Pilot studies found no significant differences between THI self-report and therapist records for number of psychotherapy hours [37]. In the current study, THI-M responses were validated against available administrative data sources (i.e., electronic health records). In the baseline assessment, the THI-M measured service utilization over the past year; in follow-up assessments, the THI-M measured service utilization since the previous assessment.

## Cost assessment

The cost of each treatment condition was calculated from a healthcare system perspective, given that the MHS is the sole payor and primary decision-maker with regard to implementation of these suicide prevention services. Total treatment costs were comprised of two categories: training and implementation activity costs and treatment delivery costs.

**Training and implementation activity costs.** Micro-costing was applied to determine the cost of training and implementation activities beyond standard clinical practice [31,38,39]. Costs related to these activities only applied to the CAMS condition in the current analyses as ETAU therapists did not receive any additional training or require preparatory activities for the provision of treatment. Further, although ETAU therapists were offered additional supervision and consultation as needed, key informant interviews indicated that they elected not to participate in these activities during the course of the study. This may reflect stronger allegiance to a specific treatment approach for managing suicidality. Consequently, there were no additional costs beyond those captured in the provision of clinical services, as described in a later section. For costs related to training and implementation activities, micro-costing entailed the following steps:

*Itemization of activities.* An inventory was developed listing all major non-clinical activities that occur for each treatment condition at the individual level, including training and other preparatory activities that must take place prior to implementing the intervention. Activities

performed were identified by study protocols, intervention manuals, administrative data systems maintained by research personnel, and qualitative interviews with research personnel and study therapists as feasible and appropriate.

*Estimating resource inputs and unit costs.* The resources used to perform each activity were identified and then the amounts of each resource used were estimated. Resources included personnel time, travel, and materials.

Costs for personnel time were measured in hourly units based on 2018 annual salary plus fringe benefits, divided by 2080 hours (an assumed 40 hours per week). Annual base salaries for CAMS trainers and consultants were estimated using the Bureau of Labor Statistics Occupational Employment Statistics Query System based on geographical area and occupation classification. Study investigators' salaries were included to reflect the real-world cost of CAMS training delivered directly by the developers of the intervention. Study therapists were clinical social workers assumed to be GS-12 paygrade based on key informant interviews; annual base salaries were estimated based on the 2018 General Schedule Base Pay Scale [40] and adjusted for locality. A fringe benefit rate of 29.1% was applied to annual base salaries for study investigators and therapists based on the 2018 national average for private industry workers in the South Atlantic region [41]. Annual base salaries and fringe rates for four pre-doctoral graduate student assistants who contributed to fidelity assessment were estimated based on typical grant-funding allowances reported by The Catholic University of America Office of Sponsored Programs.

Travel costs reflect resources needed for personnel to attend on-site trainings of CAMS study therapists. Two on-site trainings were conducted at the military installation as part of the OWL study; however, the current cost analyses assumed that only one on-site training would be required to better approximate real-world implementation not constrained by research conditions. Costing of on-site training assumed attendance by two CAMS study investigators and four study therapists. Because the OWL study was funded through a federal grant, travel costs were assumed to be consistent with General Services Administration allowable rates. Cost of airfare was estimated using the 2018 maximum allowable airfare rate determined by the General Services Administration City Pair Program [42]. Cost of lodging was estimated based on 2018 hotel rates negotiated by the federal government by locality [43]. Cost of meals and incidental expenses was estimated based on 2018 government allowable per diem rates by locality [44]. Traveling personnel shared one rental car at a cost estimated based on 2018 daily government rental car rates by locality (www.FedTravel.com).

The on-site training required facility space large enough to host all attendees in the same room. The size of the room was estimated to be approximately 300 square feet. Facility costs were estimated based on the current median cost for office space in the closest available locality [45], adjusted to reflect 2018 dollars. Specifically, the median annual cost per square foot was multiplied by 300 square feet and then divided by 2,080 hours (the total number of hours the room could be used annually). Hourly costs of electricity were estimated using 2018 commercial electricity costs in the region per the Department of Energy [46].

Costs for materials (for example, training or psychoeducational materials) were estimated per item based on study receipts, treatment protocols, and/or information obtained in key informant interviews.

*Calculating total costs.* The total cost per activity was calculated by multiplying resource inputs by per-unit costs. Costs were then summed to determine the total cost for training and non-clinical implementation activities across the CAMS condition. Because training and implementation costs were incurred at the treatment condition-level, a fixed, per-individual cost was calculated by dividing total cost of activities by the total number of individuals a trained CAMS therapist might ultimately treat with the CAMS intervention during his/her

career as an MHS provider based on rates of suicidal ideation in military populations [7,33,47–50] and estimated typical caseload size and length of career for MHS behavioral health providers [51,52]. These estimates also assume a relatively high degree of adherence to CAMS across the provider's MHS career, as supported by the absence of drift reported in the primary trial [24] and a community survey of mental health practitioners reporting generally high levels of adherence to the CAMS therapeutic approach and practice [53]. The fixed cost of non-clinical implementation activities was then added to each CAMS participant's study treatment delivery costs to estimate the overall per-individual cost of the intervention.

**Treatment delivery costs.** The cost of delivering each intervention was estimated at the individual level based on the number of treatment visits attended as documented in THI-M assessments and validated by administrative study records. Each visit was assigned an appropriate Current Procedural Terminology (CPT) [54] and costed using corresponding 2018 TRICARE/CHAMPUS Maximum Allowable Charges based on facility, non-physician provider, and appropriate locality rates. TRICARE (formerly known as CHAMPUS) is a DoD health insurance program, and its Maximum Allowable Charges rates reflect costs directly related to provision of the service (i.e., provider time used in the visit), practice expenses such as facilities and administrative staff, and malpractice insurance. This same procedure was applied to missed treatment visits (i.e., no-shows) to reflect the missed opportunity to deliver services and collect payment for dedicated provider time [55]. Out-of-session contacts that occurred as part of treatment (e.g., phone calls) were assigned a CPT code based on the purpose and duration of the communication and costed accordingly.

## Benefits assessment

Benefits, in the form of cost savings, were then estimated based on healthcare expenditures for behavioral health services, medical services, crisis services, and certain medications using the THI-M. Each visit was assigned an appropriate CPT code(s) and direct care costs for these services were estimated based on the TRICARE Reimbursement Manual and CHAMPUS Maximum Allowable Charge rates. Of note, because the nature of available data did not allow for determination of exact timing of service delivery, all non-study-treatment services were captured in the assessment of benefits rather than treatment costs, including those delivered during windows with active study treatment.

## Analyses

Outcome analysis used an intent-to-treat approach that included all participants who completed a baseline assessment. Multiple imputation using chained equations with predictive mean matching was used to minimize uncertainty related to the replacement of missing data for health services utilization [56]. Treatment cost data did not require imputation as data were available for all participants. Participants who did not complete at least one follow-up assessment ($n = 6$) were excluded from imputation and subsequent analyses. Fifteen imputed datasets were created to adequately address the observed proportion of missing data [57,58]. Planned statistical analyses were carried out on each imputed dataset, producing separate summary statistics (e.g., means or regression coefficients) with corresponding standard errors. For regression analyses, Rubin's rules [59] were applied to derive pooled coefficients and standard errors [60]. For nonparametric tests, observed and imputed datasets were analyzed and reported separately.

Descriptive analyses were conducted to examine unadjusted costs per individual, major category of activity, and treatment condition at each follow-up timepoint. General linear mixed models (LMMs) were used to assess group differences in intervention costs at each follow-up

timepoint with a random intercept and slope for participant. Wilcoxon rank-sum tests were conducted to examine differences in median costs, a useful measure of the most typical cost per individual.

Benefits were examined both between-participant (i.e., ETAU versus CAMS) and within-participant (i.e., pre- versus post-intervention). LMMs were used to assess group differences across time in total healthcare expenditures and by categories of expenditure (i.e., behavioral health, medical providers, crisis services, and medications). Linear regression was used to assess within-subjects change in pre- versus post-intervention past-year healthcare expenditures at 12-month follow-up. Cost-benefit ratios (CBRs; derived as the cumulative monetary benefit of treatment divided by the cumulative cost) and net benefit (derived as the cumulative benefit minus the cumulative cost of treatment) were calculated at the individual level. Linear regression was used to examine group differences in cost-benefit and net benefit at 12-month follow-up, as these outcomes reflect pre-intervention versus post-intervention change in past-year expenditures [61].

For all regression models of repeated-measures outcomes, treatment condition, time, and an interaction term were entered as fixed effects, with a random intercept for participants and random slope for participants by time. Nesting of participants within study therapists was examined but did not significantly improve model fit and was therefore not used in analyses. A stepwise procedure was used to identify appropriate demographic and clinical covariates as fixed effects terms, and likelihood ratio tests were used to examine random parameters. All statistical analyses were conducted using Stata v.16.0.

## Results

### Sample at baseline

Of the 148 individuals enrolled in the study, 73 were randomized to the CAMS condition and 75 were randomized to ETAU. The full study sample ($N$ = 148) was predominantly male (80%), White (53%), and married (51%), with a mean age of 26.8 ($SD$ = 5.9). Participants mostly held the rank of junior enlisted (E1-E4; 70%). The mean total SSI-C score at baseline was 20 ($SD$ = 5.3), and half of participants reported a lifetime history of at least one suicide attempt, with 27% reporting multiple attempts. There were no statistically significant differences between the CAMS and ETAU conditions with regard to patient participants' sociodemographic or baseline clinical characteristics.

### Missing data

Study retention rates in the intent-to-treat sample were 96%, 89%, 79%, and 78% for CAMS and 90%, 83%, 79%, and 77% for ETAU at 1-, 3-, 6-, and 12-month follow-ups, respectively, with an overall missing data rate of 13% for primary outcomes. Separate mixed effect logistic regression models revealed no statistically significant differences in rates of missing data across time based on age, gender, ethnicity, marital status, education level, rank, number of lifetime suicide attempts, treatment condition, or primary study therapist.

### Costs

The total cost of CAMS training and consultation activities was $7,960.32; assuming each of the four CAMS clinicians could treat 160 individuals over the course of an MHS career, this amounted to a cost of $12.44 per participant. Table 1 displays the results of micro-costing procedures for these activities [31].

**Table 1. Resource x activity table for CAMS training and consultation activities.**

| Resource | Unit Measure | Units Required | Total Resource Cost |
|---|---|---|---|
| **Review CAMS Manual** | | | |
| *Time* | | | |
| Clinicians[a] | 1 hour | 13 | $592.02 |
| *Materials* | 1 manual | 4 | $152.04 |
| **Attend On-site Training** | | | |
| *Time* | | | |
| Clinicians[a] | 1 hour | 48 | $2,185.92 |
| Study Investigators | 1 hour | 24 | $1,888.32 |
| *Facilities* | | | |
| Office Space | 1 hour | 12 | $23.37 |
| Electricity | 1 kW hour | 12 | $1.14 |
| *Materials* | 1 page | 80 | $16.00 |
| *Travel* | | | |
| Airfare | 1 roundtrip ticket | 2 | $944.00 |
| Lodging | 1 room/night | 4 | $372.00 |
| Rental Car | 1 day | 2 | $130.00 |
| Meals & Incidental Expenses Per Diem | 1 day | 6 | $255.00 |
| **Case Consultation[b]** | | | |
| *Time* | | | |
| Clinicians[a] | 1 hour | 16 | $728.64 |
| Study Investigators | 1 hour | 12 | $671.88 |
| **TOTAL** | | | **$7,960.32** |

[a] Costs reflect training and consultation activities for four CAMS clinicians.

[b] Assumes a total of four hourly group consultation calls, as is typical in real-world CAMS implementation training.

CAMS−Collaborative Assessment and Management of Suicidality; kW−kilowatt.

Table 2 shows descriptive statistics for cumulative costs of treatment over time by treatment condition. Mean total study treatment cost at 12-month follow-up was $434.82 ($SD$ = $209.66) for ETAU and $433.38 ($SD$ = $248.29) for CAMS; LMM results indicated that treatment conditions did not differ significantly in treatment costs across time. Wilcoxon rank-sum tests examining group differences in median total study treatment costs were consistent with linear mixed modeling (p = .57 to .80). The average number of active study treatment sessions attended at 12 months was comparable between treatment conditions ($M$ = 6.40 [$SD$ = 3.53] for ETAU, $M$ = 6.20 [$SD$ = 3.89] for CAMS; $Z$ = 0.69, $p$ = .49), as was the average number of out-of-session contacts ($M$ = 0.56 [$SD$ = 1.27] for ETAU, $M$ = 0.59 [$SD$ = 1.67] for CAMS; $Z$ =

**Table 2. Descriptive statistics for total cumulative study treatment costs.**

| Timepoint | Mean (*SD*) | | Median (95% *CI*) | |
|---|---|---|---|---|
| | **ETAU** | **CAMS** | **ETAU** | **CAMS** |
| **1-month** | $282.64 ($86.46) | $273.55 ($75.17) | $247.92 ($247.92-$309.9) | $260.35 ($260.35-$322.33) |
| **3-months** | $404.53 ($168.94) | $393.58 ($138.74) | $371.88 ($309.90-$433.86) | $384.31 ($322.33-$384.31) |
| **6-months** | $434.82 ($209.66) | $427.84 ($221.70) | $371.88 ($309.90-$448.55) | $384.31 ($322.33-$384.31) |
| **12-months** | $434.82 ($209.66) | $433.38 ($248.38) | $371.88 ($309.90-$448.55) | $384.31 ($322.33-$384.31) |

SD−standard deviation; CI−confidence interval; ETAU−enhanced treatment as usual; CAMS−Collaborative Assessment and Management of Suicidality.

-0.37, $p$ = .71) and average number of missed visits ($M$ = 0.55 [$SD$ = 1.33] for ETAU, $M$ = 0.81 [$SD$ = 2.20] for CAMS; $Z$ = -0.97, $p$ = .33).

## Benefits

Due to a few extreme outliers, crisis services and total healthcare expenditure variables were separately Winsorized at the 1st and 99th percentiles to better approximate a normal distribution [62]. Table 3 presents descriptive statistics for cumulative healthcare expenditures for each treatment condition by category of service and timepoint. At baseline, mean past year total healthcare expenditures were comparable between the two treatment conditions after controlling for gender and race ($p$ = .17). Multivariate LMMs of cumulative total healthcare expenditures across follow-up timepoints revealed a significant interaction of treatment condition x time at 6-months (β = -2,480.34, $SE$ = 1,198.95; $p$ = .04), indicating a decreased rate of healthcare expenditure for the CAMS condition at this timepoint.

In models of cumulative healthcare expenditures by category of services, there were no significant differences by treatment condition with respect to behavioral health services, medical provider services, or medications. However, for crisis services, a significant interaction of treatment condition x time was found at 6-month (β = -2,036.12, $SE$ = 793.38; $p$ = .01) and 12-month (β = -2,743.24, $SE$ = 1,300.88; $p$ = .04) follow-ups, indicating a decreased rate of crisis services expenditures for the CAMS condition.

In examining within-subject effects, mean changes in pre- versus post-intervention total past-year healthcare expenditures suggested a slight reduction in past-year expenditures for CAMS participants at 12-month follow-up ($M$ = -$14.21, $SD$ = $13,075.90), whereas past-year expenditures increased for ETAU participants ($M$ = $3,304.72, $SD$ = $14,756.44). Aggregated across participants by treatment condition, this represents a pre- versus post-intervention increase in total past-year healthcare expenditures of $234,637.59 for ETAU compared to a *decrease* of $1,009.42 for CAMS. Still, linear regression indicated that within-subject changes in pre- to post-intervention expenditures were not significantly different by treatment condition, likely due to substantial variability among observations.

## Cost-benefit

Mean CBRs at 12-month follow up were -13.26 ($SD$ = 55.10) for ETAU and 1.68 ($SD$ = 40.53) for CAMS. Linear regression revealed that CAMS participants had significantly greater CBRs

**Table 3. Descriptive statistics for cumulative healthcare expenditures.**

| Timepoint | Behavioral Health $M(SD)$ | Medical Providers $M(SD)$ | Crisis Services $M(SD)$ | Medications $M(SD)$ | Total $M(SD)$ |
|---|---|---|---|---|---|
| **1-month** | | | | | |
| ETAU | $344.56 ($733.37) | $445.57 ($339.55) | $828.19 ($2,397.55) | $58.35 ($68.60) | $1,676.68 ($2,535.78) |
| CAMS | $394.28 ($409.52) | $373.07 ($312.23) | $1,567.83 ($3,588.47) | $48.15 ($56.80) | $2,385.66 ($3,881.25) |
| **3-months** | | | | | |
| ETAU | $846.53 ($1,960.07) | $866.41 ($650.71) | $2,319.58 ($4,584.65) | $157.11 ($183.43) | $4,035.30 ($5,228.14) |
| CAMS | $739.42 ($630.30) | $923.15 ($689.72) | $2,686.91 ($5,071.78) | $122.11 ($143.79) | $4,471.59 ($5,835.28) |
| **6-months** | | | | | |
| ETAU | $1,587.44 ($3,005.15) | $1,445.01 ($974.37) | $4,882.80 ($7,167.52) | $307.61 ($339.82) | $8,433.70 ($9,730.96) |
| CAMS | $1,412.42 ($2,508.25) | $1,530.42 ($1,148.55) | $3,879.50 ($6,241.34) | $232.08 ($310.09) | $7,054.42 ($7,660.61) |
| **12-months** | | | | | |
| ETAU | $2,693.53 ($5,425.58) | $2,384.50 ($1,589.13) | $7,386.36 ($9,223.68) | $608.02 ($641.90) | $13,256.40 ($14,094.71) |
| CAMS | $2,569.90 ($5,226.55) | $2,395.66 ($1,929.21) | $5,576.90 ($7,286.21) | $446.07 ($566.48) | $11,004.54 ($10,388.93) |

M−mean; SD−standard deviation; ETAU−enhanced treatment as usual; CAMS−Collaborative Assessment and Management of Suicidality.

(i.e., more benefit per dollar spent) compared to ETAU condition after controlling for race and gender (β = 21.67, $SE$ = 8.17; $p$ < .01). Mean within-subject net benefit at 12-months was -$3,739.54 ($SD$ = $14,755.23) for ETAU and -$419.13 ($SD$ = $12,941.40) for CAMS, indicating that descriptively neither treatment paid for itself through reduced healthcare expenditures. However, across all datasets, one-sample Wilcoxon signed-rank tests showed that median net benefits were significantly less than zero for ETAU whereas they were not significantly different from zero for CAMS. Further, linear regression revealed that CAMS was associated with significantly greater net benefit compared to ETAU after controlling for gender and race (β = -5,167.93, $SE$ = 2,271.79; $p$ = .03).

## Discussion

This study examined the costs, benefits, and cost-benefit of CAMS versus ETAU for the treatment of suicidality in active duty service members from a healthcare perspective. Consistent with a preliminary cost analysis [16], there were no significant differences in costs of treatment between CAMS and ETAU. As anticipated, the CAMS intervention did require training and consultation activities over and above that of usual care; however, the costs of these activities were relatively minimal when spread across the entire population of beneficiaries.

With regard to benefits at the group level, CAMS was associated with significantly reduced total cumulative healthcare expenditure compared to ETAU at 6-month follow-up, likely driven by significantly lower crisis services expenditures. However, ETAU eventually matched CAMS in total cumulative healthcare expenditures at 12-month follow-up timepoints, although crisis services expenditures remained significantly lower for CAMS. The data suggest that participants in the CAMS condition may use more crisis services than ETAU in the first month of treatment but use significantly fewer crisis services by 6- and 12-month follow-ups. The early uptick in crisis service utilization may be explained by CAMS's suicide-specific focus which promotes accessing such services in an acute crisis, whereas the decreased utilization by six months likely reflects findings from the primary trial that CAMS participants had a lower probability of having suicidal ideation at 3-month follow-up [24]. In examining within-subject pre- versus post-intervention past-year healthcare expenditures, no significant differences were found between CAMS and ETAU participants for any category of healthcare expenditures.

CBRs were significantly greater for CAMS participants at 12-month follow-up, indicating greater benefit per dollar spent compared to ETAU. On average, CAMS participants reduced their past-year total healthcare expenditures by $0.68 for each dollar spent on treatment, whereas ETAU participants increased spending by $13.26 for each dollar spent on treatment. Neither treatment condition evidenced positive net benefit; that is, neither treatment paid for itself through reduced healthcare expenditures. Still, net benefit was significantly greater in the CAMS condition compared to ETAU by approximately $3,320 per individual, on average.

### Limitations and future directions

This study had several limitations. With regard to estimating intervention costs, our micro-costing approach likely resulted in inflated estimates of CAMS costs. Micro-costing was used in the current study to maximize precision in cost estimates; however, a known limitation of micro-costing is a potential lack of generalizability beyond the specific setting or population being studied [63]. Because the current analyses were based on data from a randomized controlled trial designed to ensure internal validity, greater resources were required to provide CAMS than would be observed in real-world settings. We attempted to mitigate potential overestimation of costs related to research procedures through application of some real-world

practice assumptions to enhance external validity. Still, costs of real-world CAMS delivery may be substantially lower in some settings owing to the availability of less expensive training formats (e.g., online video courses; www.CAMS-care.com) and delivery formats (e.g., group therapy) [64], as well as its suitability for use by diverse types of providers (e.g., paraprofessionals) [13]. Conversely, costs may be greater in settings where outpatient mental health services are typically delivered by doctoral-level therapists. Additional cost-inclusive evaluations of CAMS across these various implementation approaches are needed to inform wider dissemination of the framework.

A second limitation of the current study is that is solely reflects the MHS perspective. While our findings may inform decision-making that maximizes impact for the payer, it is important to consider that such a narrow scope likely undermines the true benefit of an intervention [65]. Future cost-inclusive analyses of CAMS should consider adopting a broader range of perspectives, particularly a societal perspective in which all costs and outcomes associated with an intervention are taken into consideration regardless of whom experiences them directly [66,67]. For example, it is possible that effective treatment of suicidality may also produce improvements related to productivity, employment, crime, incarceration, and consumption that further enhance societal welfare [68]. Similar improvements may also extend to others impacted by the individual's suicidality (e.g., family members, friends, colleagues, and clinicians). Indeed, one study found that indirect costs represented 97.1% of the $58.4 billion total U.S. economic costs of reported suicide-related acts in 2013 [69]. In light of research indicating that Soldiers may be more likely to attempt suicide if there has been a past-year suicide attempt within their unit [70], it is conceivable that effective treatment of suicidality might even have a multiplicative effect within a military population. Incorporating these broader societal considerations into future cost-inclusive evaluations of suicide interventions may enhance our understanding of their true value.

Third, in the original trial, study clinicians were assigned to their respective treatment conditions based on their preferred approach for managing suicidal patients. Although this approach maximized fidelity to study treatment, it should be noted that CAMS therapists inadvertently had more than twice the years of practice experience since professional degree than ETAU therapists. This finding is consistent with research suggesting that less-experienced therapists may be more inclined to adhere to a specific treatment protocol or newly learned therapy whereas more seasoned providers may be more open to a variety of approaches [71–73]. Previous research suggests that years of practice experience is not necessarily correlated with therapist effectiveness [74,75] and statistical modeling in the current study did not suggest any therapist effects on clinical outcomes. Further, differences in practice experience did not impact treatment costs between treatment conditions as all study therapists were classified as "non-physician" under TRICARE reimbursement procedures [76]. Still, some studies indicate that greater suicide-specific practice experience is associated with enhanced suicide intervention competence and skills [77,78], and thus future CAMS investigations would be strengthened by ensuring comparable practice experience among providers to remove this potential confounding.

Finally, due to limitations in the available data, the current study was unable to determine the exact temporal sequence of healthcare services delivered to participants within each follow-up window. It is therefore possible that some non-study-related services classified as outcomes (i.e., benefit) were actually delivered concurrently with active study treatment rather than post-treatment, as the designation of outcome would imply. Although this approach allowed for the greatest use of service utilization data while avoiding conflation with study treatment costs, future cost-inclusive evaluations would benefit from greater precision in the assessment of benefits.

## Conclusions

Suicide prevention represents a top force health and readiness priority for the DoD. The CAMS framework has been increasingly recognized as a promising approach for the treatment of suicidal risk. Although CAMS has amassed a robust evidence base supporting its clinical effectiveness, the current study provides evidence that CAMS may also be associated with greater benefit and cost-benefit compared to treatment as usual. Although the differences observed in our study were small, the potential impact grows appreciably once extrapolated across the population of 1.38 million active duty service members (and 9.5 million beneficiaries total) served by the MHS [79]. These economic advantages may become more pronounced as CAMS's more cost-effective training and delivery formats become more widely adopted. Ideally, such cost savings could allow limited financial resources to be re-allocated toward meeting other healthcare system needs. Although additional evaluations are necessary to enhance the generalizability of our findings, the current study provides a valuable insight into the costs and benefits of CAMS within the context of the MHS. These findings may inform more robust cost-inclusive analyses in future trials and ultimately facilitate decision-making among the policymakers, healthcare administrators, and clinicians tasked with selecting mental health programs that will provide the biggest 'bang for the buck' amidst constrained resources.

## Author Contributions

**Conceptualization:** Phoebe K. McCutchan, Brian T. Yates, David A. Jobes, Amanda H. Kerbrat, Katherine Anne Comtois.

**Data curation:** Phoebe K. McCutchan, Amanda H. Kerbrat.

**Formal analysis:** Phoebe K. McCutchan.

**Investigation:** Phoebe K. McCutchan, David A. Jobes, Amanda H. Kerbrat, Katherine Anne Comtois.

**Methodology:** Phoebe K. McCutchan, Brian T. Yates, David A. Jobes, Amanda H. Kerbrat, Katherine Anne Comtois.

**Project administration:** Phoebe K. McCutchan, David A. Jobes, Amanda H. Kerbrat, Katherine Anne Comtois.

**Resources:** Phoebe K. McCutchan, David A. Jobes, Amanda H. Kerbrat, Katherine Anne Comtois.

**Supervision:** Brian T. Yates, David A. Jobes, Katherine Anne Comtois.

**Validation:** Phoebe K. McCutchan.

**Visualization:** Phoebe K. McCutchan, Amanda H. Kerbrat.

**Writing – original draft:** Phoebe K. McCutchan, Brian T. Yates, David A. Jobes.

**Writing – review & editing:** Phoebe K. McCutchan, Brian T. Yates, David A. Jobes, Amanda H. Kerbrat, Katherine Anne Comtois.

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
