## [Decision Letter · Decision Letter 0]

11 Jun 2021

PONE-D-21-12016

Costs, Benefits, and Cost-Benefit of Collaborative Assessment and Management of Suicidality versus Enhanced Treatment as Usual

PLOS ONE

Dear Dr. McCutchan,

Thank you for submitting your manuscript to PLOS ONE. After careful consideration, we feel that it has merit but does not fully meet PLOS ONE’s publication criteria as it currently stands. Therefore, we invite you to submit a revised version of the manuscript that addresses the points raised during the review process.

We look forward to receiving your revised manuscript.

Kind regards,

Isabelle Durand-Zaleski

Academic Editor

PLOS ONE

Journal Requirements:

"I have read the journal's policy and the authors of this manuscript have the following competing interests: David Jobes has conflicts to disclose related to grant funding from the National Institute of Mental Health; he receives book royalties from the American Psychological  Association Press and Guilford Press; and he is the founder of CAMS-care, LLC (a professional training and consultation company). Phoebe McCutchan, Brian Yates, Amanda Kerbrat, and Katherine Comtois have declared that no competing interests exist."

Reviewers' comments:

Reviewer's Responses to Questions

**Comments to the Author**

1. Is the manuscript technically sound, and do the data support the conclusions?

Reviewer #1: Yes

Reviewer #2: Yes

2. Has the statistical analysis been performed appropriately and rigorously? 

Reviewer #1: I Don't Know

Reviewer #2: Yes

3. Have the authors made all data underlying the findings in their manuscript fully available?

Reviewer #1: Yes

Reviewer #2: Yes

4. Is the manuscript presented in an intelligible fashion and written in standard English?

Reviewer #1: Yes

Reviewer #2: Yes

5. Review Comments to the Author

Reviewer #1: Article well written and well presented. I have only a few remarks to propose.

This article is focused on a medico-economic analysis of a prevention program (CAMS) on suicidal behaviors in suicidal soldiers of the US army. This intervention is delivered by social workers, and is compared to an Enhanced version of Treatment As Usual ETAU).

I would be interested if some items could be further discussed:

1) Social workers are not randomized and in CAMS program are far more experienced. It is only discussed in the limitations part saying it has no influence. Some articles emphasize that experience might be important. (For instance: Suicide intervention skills and related factors in community and health professionals Suicide Life Threat Behav . 2010 Apr;40(2):115-24. doi: 10.1521/suli.2010.40.2.115.)

2) CAMS protocol stops if 3 consultations in a row are without suicidal ideas, as ETAU does not. This might have direct economic implications.

3) It seems that Crisis Services use is different in the 2 programs and seems more important in CAMS program in the 1st month, and less at 6 months and 12 months. This seems very interesting and should be further discussed.

Reviewer #2: Thank you for the opportunity to review this interesting paper. Similarly to reviewer 2, I was not among the initial reviewers and carried out a first blind review of the manuscript before considering the authors’ responses to the initial review and revising my recommendations based on things that had already been addressed or discussed (which contributes to the reduced number of comments I have). Overall, I believe that this article is of research and policy interest, rigorous, well-written and clear. I, however, still have a couple of comments to improve the latest version of the manuscript but want to stress that in my view the comments of previous reviewers have been appropriately addressed overall.

1/ I believe that, for non-US readers, some parts could be made clearer or more specific. In particular, behavioral health is a concept which is very specific to the US and which would be worth defining once. The involvement of social workers in mental care is also very specific to your national context and could require some additional information to justify why you used social workers wages in your economic evaluation. In the introduction, it would be interesting to provide more specific figures on the unprecedented rate of increase in suicides among the military workforce in the US over time since such figures appear to be available. Also briefly mention what is TRICARE and CHAMPUS. Minor additional precisions needed: “Active component”: clarify for readers what this refers to; “condition”: maybe replace everywhere by “treatment condition” as you do once, that way it is less confusing for non-native English speakers (as conditions can refer to diseases); “modeling on multiply imputed datasets” (abstract): multiple?, “member of the Warriors in Transition unit”: explain what it is very briefly?

2/ I think the discussion or conclusion could stress more the policy implications of the research for the military health system (make it more explicit maybe).

3/ I am surprised that none of the ETAU clinicians, when provided with potential supervision upon request, asked for it. Maybe discuss potential hypotheses about why this happened (while it seems to be less the case with CAMS while it is carried out by providers with more years of experience).

4/ At some point you mention the time required for transit between the patient’s place of residence and the place of care and justify that you did not include it in your economic evaluation because you were not able to access confidential information regarding the place of residence of patients. I wonder if anyway this cost should be included in an evaluation adopting the military health services perspective? Would not those costs lie on the patients anyway or be more related to loss of productivity for the military?

5/ I read your answer to a previous comment on the pre-doctoral graduate students included in the cost assessment. I am still not fully convinced they should be included in an evaluation which aims to accurately fit real-world settings. “We presume these cost rates are comparable to those of staff who might conduct administrative activities in real-world settings.”: This should be more justified to be convincing.

6/ Some reorganization might be necessary: “In the current secondary analysis, treatment costs […] were determined using micro-costing” and subsequent paragraph of the introduction: this would be better suited in the method section; “Multiple imputation using chained equations with predictive mean matching was used to minimize uncertainty…”: this would be better suited in the method section of the paper as well.

7/ For the ethical approvals that you mention in the first paragraph of the method section, it would better (more transparent and traceable) to also provide the identification number of the approval received.

8/ Measures section: “were validated against available administrative data sources (e.g., electronic health records”: it would be better to be exhaustive and more specific here; and the comma after e.g. (or sometimes i.e. in the text) seems misplaced. Similarly, in the ‘Training and implementation activity costs’ paragraph: “Activities performed were identified by intervention manuals, administrative data systems, and qualitative interviews as feasible and appropriate”: it would be better to be more precise here.

9/ “There were no statistically significant differences between the CAMS and ETAU conditions with regard to sociodemographic or baseline clinical characteristics”: maybe specify that you are talking about patients’ characteristics.

10/ I must admit I am a bit concerned about the involvement of the founder of a company called CAMS-care in a research which strongly supports the overall value of CAMS.

6. PLOS authors have the option to publish the peer review history of their article (what does this mean?). If published, this will include your full peer review and any attached files.

Reviewer #1: No

Reviewer #2: No

---

## [Author Response · Author response to Decision Letter 0]

26 Jul 2021

Dear Dr. Robinson, Dr. Sisask, Dr. Kõlves, Dr. Oquendo, Dr. Heber, and Reviewers,

Thank you for your review of our manuscript entitled, “Costs, Benefits, and Cost-Benefit of Collaborative Assessment and Management of Suicidality versus Enhanced Treatment as Usual” (PONE-D-20-33804). We are pleased to respond to each Journal and Reviewer comment below.

Journal requirements: When submitting your revision, we need you to address these additional requirements.

We have reviewed the style requirements and believe our submission to be consistent with the guidance provided.

2. Thank you for stating the following in the Competing Interest Section:

"I have read the journal's policy and the authors of this manuscript have the following competing interests: David Jobes has conflicts to disclose related to grant funding from the National Institute of Mental Health and the American Foundation for Suicide Prevention; he receives book royalties from the American Psychological Association Press and Guilford Press; and he is the founder of CAMS-care, LLC (a professional training and consultation company). Phoebe McCutchan and Brian Yates have declared that no competing interests exist."

2.1. We note that you have provided funding information that is not currently declared in your Funding Statement. Currently, your Funding Statement reads as follows:

The current study represents an unfunded secondary analysis of a randomized-controlled trial. The primary trial was supported by the Department of the Army through federal grant W81XWH-11-1-0164, awarded and administered by the Military Operational Medicine Research Program (MOMRP). We have added this information to the cover letter for clarity. Funding described in Dr. Jobes’ conflicts of interest disclosure did not apply to the current study. 

2.2. Please confirm that this ("he receives book royalties from the American Psychological Association Press and Guilford Press; and he is the founder of CAMS-care, LLC (a professional training and consultation company)") does not alter your adherence to all PLOS ONE policies on sharing data and materials, by including the following statement: "This does not alter our adherence to PLOS ONE policies on sharing data and materials.” (As detailed online in our guide for authors http://journals.plos.org/plosone/s/competing-interests). If there are restrictions on sharing of data and/or materials, please state these. Please note that we cannot proceed with consideration of your article until this information has been declared.

The authors confirm that Dr. Jobes’s competing interests do not alter our adherence to all PLOS ONE policies on sharing data and materials. We have included the statement, “This does not alter our adherence to PLOS ONE policies on sharing data and materials” in our revised cover letter. 

Please note that we have made an additional change was made to the Competing Interests statement in the cover letter. Dr. Jobes no longer receives grant funding from the American Foundation for Suicide Prevention; the statement has been updated accordingly.

The dataset uploaded as a Supporting Information file in our original submission has been deleted to ensure compliance with IRB requirements. We have added a Data Availability Statement detailing the restrictions and providing contact information for the institutional body to which data requests may be sent. Because no Supporting Information files are included as part of the current submission, no captions were added to the revised manuscript.

Reviewer #1 comments:

Reviewer's Responses to Questions

Comments to the Author

1. Is the manuscript technically sound, and do the data support the conclusions?

Reviewer #1: Partly

2. Has the statistical analysis been performed appropriately and rigorously?

Reviewer #1: Yes

3. Have the authors made all data underlying the findings in their manuscript fully available?

Reviewer #1: Yes

4. Is the manuscript presented in an intelligible fashion and written in standard English?

Reviewer #1: Yes

5. Review Comments to the Author

Reviewer #1: The authors examined the economic cost-benefit ratio of CAMS versus enhanced treatment-as-usual (TAU) in a sample of 148 active duty military with current suicidal ideation. The study involved a secondary analysis of data from a randomized clinical trial of CAMS versus TAU for soldiers with current suicidal ideation who obtained services at outpatient mental health clinics. The study addresses a topic of substantial importance. The suicide rate is rising, and we have limited information available about the costs of treatment, which are critical to decision making and actual implementation. The study and manuscript also have several strengths, particularly in the description of treatment training and implementation costs, and in the description of data analyses. However, in some areas the manuscript is somewhat vague and the presentation of background information, findings, and their interpretation would benefit from being more balanced (critical) and scholarly. A clear statement of hypotheses and primary outcomes would also be helpful.

The following suggestions may enable the authors to improve their manuscript.

1. Abstract: It is recommended that the statement be removed that CAMS is “one of the most empirically-supported suicide-focused interventions to date” and has “demonstrated clinical effectiveness across a number of clinical trials.” This seems too broad sweeping for this abstract, particularly as it’s noted in Introduction that the need for studies with more robust designs has been emphasized.

These statements have been removed from the Abstract.

The phrase “to examine the economic costs, benefits, and cost-benefit” is confusing as used in the Abstract and throughout the manuscript. Is there a way to put this more succinctly and define the term(s) in Introduction? It seems to be a “cost analysis” that considers financial costs and Also, the differential meanings of “cost-benefit” and “net benefit” are unclear in the Abstract.

We appreciate the Reviewer highlighting that these economic terms may be confusing for readers. The overarching aim of the study was, as stated, to examine the economic costs, benefits, and cost-benefit of the interventions (that is, the costs of delivering treatment, the value of resources generated or saved as a result of the intervention, and measures of costs relative to resources gained). We view these as three distinct economic constructs and have respectfully opted not to revise this statement in the Abstract; however, the Introduction was revised to operationalize these terms and provide greater clarity on their distinctions as primary outcomes. 

We also recognize that the distinction between ‘cost-benefit’ and ‘net-benefit’ is unclear in the Abstract. We have revised the Abstract to reflect that the construct of ‘cost-benefit’ was assessed using two distinct indices common in economic evaluation, cost-benefit ratios (i.e., ratios of benefits divided by costs) and net benefit (i.e., benefits minus costs): “Cost-benefit was examined in the form of cost-benefit ratios and net benefit.” We have also revised the Introduction to include clarification of these as outcome variables.

It would be helpful to include the sample size and a brief statement of demographics.

The Abstract was revised to include the following statement, “The full intent-to-treat sample included 148 participants (mean age 26.8 years± 5.9 SD years, 80% male, 53% White).”

2. Introduction: As the suicide rate has increased in all age groups in recent years, it would be helpful to state the 2018 rate for the age and gender-adjusted U.S. general population. This will enable readers to have a sense of how different it is from the rate for active military.

To provide greater clarity to this statement, we have revised it to read, “These trends are particularly alarming in that they represent the first time in recorded history that the U.S. military suicide rate has equaled or exceeded that of comparable U.S. general population cohorts, beginning in 2008 and continuing to date. According to the most recent publication of the DoD Annual Suicide Report, in 2019 the suicide mortality rate for the Active Component was statistically indistinguishable from the 2018 U.S. population rate after adjusting for age and gender [4].” 

In the second paragraph, the last sentence could also include that research is needed to carefully evaluate the effectiveness of these interventions in rigorous studies that are powered to examine key outcomes. 

The paragraph has been revised to read, “New interventions aimed specifically at managing suicidality are emerging, but additional research is needed to carefully evaluate the effectiveness of these interventions in rigorous, well-powered studies in order to inform clinical practice guidelines and facilitate integration into routine clinical care.”

Related to this, in the third paragraph, it is noted that CAMS has amassed a substantial evidence base for its effectiveness; however, information is not included about 1) the results of the “parent” RCT study (from same data set); 

The “parent” RCT study is listed as part of the citation related to the RCT evidence-base, but we appreciate how it could be more clearly referenced. Additional detail has been added to paragraph 6 of the Introduction: “The purpose of the current study was to extend previous analyses by examining the costs, benefits, and cost-benefit of CAMS compared to enhanced treatment as usual (ETAU) for the treatment of suicidality in active duty Soldiers using data from a recent clinical effectiveness RCT. Results of the primary RCT indicated that CAMS treatment was associated with significant postbaseline improvements in suicidal ideation and suicide attempt behaviors across one year of follow-up. However, the ETAU condition evidenced comparable improvements, with the exception of a significantly greater reduction in probability of suicidal ideation at 3-month follow-up observed in the CAMS condition [24]. Examination of treatment costs and benefits becomes especially useful in such scenarios where two treatment alternatives appear similarly effective, to select the intervention which maximizes return on investment.”

and 2) the limitations of research to date on CAMS and why more “robust” or rigorous studies have been recommended. These are both critical gaps in the Introduction. 

The paragraph in question has been revised to read, “Further, recent systematic review of CAMS reported it to be a promising approach to managing suicide risk and deliberate self-harm in adults [27], although it noted a high degree of heterogeneity and attrition across existing studies and emphasized the need for additional high-quality studies, particularly randomized-controlled trials (RCTs) and meta-analyses.” 

Additionally, we have added a citation for a forthcoming meta-analysis indicating that CAMS is a well-supported intervention for suicidal ideation: “Further, a more recent meta-analysis reported that CAMS treatment resulted in significantly lower suicidal ideation and general distress compared to alternative interventions, concluding that it is a well-supported intervention for suicidal ideation according to Center of Disease Control and Prevention criteria [28].”

The next paragraph (first full paragraph on second page of the Introduction) includes the statement, “…what remains to be studied is the cost of the intervention.” This is contrary to the earlier statement that more robust studies of effectiveness are still needed.

This statement has been revised to read, “In addition to its clinical effectiveness, another important factor requiring further examination is the cost of the CAMS intervention, particularly in comparison to alternatives.”

The paragraph that begins “Preliminary analyses using archival data…” seems to go too far in extrapolating from the data given that this was not a RCT and was a small study. It’s important to add a statement such as “This study involved XXX patients (XX/group) and there may have been biases in assignment to groups such that we cannot conclude that CAMS resulted in a cost savings…”

The following statement has been added, “Still, the significant methodological limitations of the study must be noted, including a small and unequally distributed sample size (N = 55; n = 25 for CAMS and n = 30 for treatment as usual), lack of random assignment, and lack of fidelity measures. Consequently, it is premature to definitively conclude based on these analyses that CAMS is associated with economic advantages compared to treatment alternatives.”

As noted above regarding the Abstract, the phrase “…the costs, benefits, and cost-benefit” would benefit from simplification and an operational definition of the one or two constructs that are primary to (related to the hypotheses of) this study.

Again, we appreciate the Reviewer’s feedback regarding the clarity of these terms for readers. The paragraph has been revised to include operationalization of each primary construct (cost, benefit, and cost-benefit): “In these analyses, treatment costs (i.e., the value of resources required to implement the intervention) were determined using micro-costing and gross-costing techniques. Benefits of the intervention (i.e., the value of resources generated or saved as a result of the intervention) were assessed as potential cost savings based on healthcare expenditures across multiple categories of services. Finally, cost-benefit (i.e., the relationship between the value of resources used and the value of resources saved by an intervention) was examined using cost-benefit ratio and net benefit metrics [31].”

Please add a clear statement of hypotheses and the study’s primary outcome(s).

The final two paragraphs of the Introduction have been revised to specifically label the five study hypotheses.

3. Materials and Methods.

How was TAU enhanced?

Paragraph 2 under Methods/Treatments states, “This condition was considered ‘enhanced’ treatment as usual in that it specified a minimum number of four treatment sessions, providers had access to consultation and supervision as needed, and Soldiers were routinely engaged and evaluated at the study assessment intervals.” We have added a citation for the primary aims manuscript (Jobes et al., 2017) which also defines the ETAU condition, and are happy to provide further clarification if necessary.

In the Measures section, it would be helpful to provide a 1-2 sentence description of the Scale for Suicidal Ideation-Current and to provide more information about the THI-M (e.g., number of items). How were follow-up assessments combined when some were missing?

A brief description of the SSI-C was added to the Measures section: “The Scale for Suicide Ideation-Current (SSI-C) [36], a 19-item interviewer-rated measure assessing suicidal ideation at its highest intensity over the past two weeks, was used to assess suicidal ideation severity (total score of 0-38) and resolution (i.e., a total score of zero). The SSI-C has demonstrated good convergent and criterion validity in previous research in psychiatric patients [36] and evidenced high internal consistency in the current study (α = .88).”

Additionally, the following details were added to the description of the THI-M: “The THI-M is an interviewer-administered measure that captures the history (e.g., service frequency, duration, and provider type) of outpatient psychotherapy and counseling visits, crisis medical services, medical provider visits, and medication use.” 

Missing data in follow-up assessments were handled using multiple imputation, as described in the Results/Missing Data Section: “Multiple imputation using chained equations with predictive mean matching was used to minimize uncertainty related to the replacement of missing data for health services utilization [55]. Treatment cost data did not require imputation as data were available for all participants. Participants who did not complete at least one follow-up assessment (n = 6) were excluded from imputation and subsequent analyses. Fifteen imputed datasets were created to adequately address the observed proportion of missing data [56, 57]. Planned statistical analyses were carried out on each imputed dataset, producing separate summary statistics (e.g., means or regression coefficients) with corresponding standard errors. For regression analyses, Rubin’s rules [58] were applied to derive pooled coefficients and standard errors [59]. For nonparametric tests, observed and imputed datasets were analyzed and reported separately.” We are happy to provide further detail if necessary.

Regarding clinicians, what was the extent of booster sessions, consultation, or feedback sessions for CAMS? For ETAU? Were these figured in?

In the effectiveness trial, CAMS therapists engaged in regular study-related consultation calls. 

Although these activities may have impacted CAMS therapists’ practice, these calls were more reflective of research procedure than real-world clinical practice. Subsequently, in an effort to represent the potential impact of these activities while avoiding unnecessarily inflating CAMS costs, we assumed that CAMS providers attended a total of four one-hour group consultation calls, as is typical in real-world CAMS-Care training. These consultation calls were factored into the costs of Training/Implementation activities, as noted in Table 1. No other booster or feedback sessions were utilized by these therapists. 

ETAU therapists were offered additional supervision/consultation by study investigators but declined usage of such resources. As stated in the Methods/Cost Assessment/Training and implementation activity costs section, “Further, although ETAU therapists were offered additional supervision and consultation as needed, key informant interviews indicated that they elected not to participate in these activities during the course of the study. Consequently, there were no additional costs beyond those captured in the provision of clinical services.”

It would be helpful to read, more specifically, about why the authors think their CAMS group was more experienced – what was pathway to becoming a CAMS therapist?

Eligible clinicians were licensed behavioral health clinicians employed by the FSGA Soldier Resiliency Center who were willing and able to consent to see randomized client participants using the treatment to which they are assigned and to consent to confidential digital recordings of treatment sessions and completion of therapist assessments. Clinicians were excluded if they were expected to leave the FSGA Soldier Resiliency Center in less than 12 months or were determined by FSGA Soldier Resiliency Center leadership to be inappropriate to participate in the study. As described in the revised Methods/Participants section, “Eligible clinicians were assigned to either the CAMS or ETAU condition based on their relative self-reported allegiance to treatment models for managing suicidal patients; clinicians reporting weaker preference for a specific treatment were assigned to the CAMS condition to ensure high adherence of ETAU clinicians to their existing approach.” 

We note in the Discussion section the observed discrepancy in practice experience as a study limitation and have revised the Discussion/Limitations section to offer the following explanation: “Third, in the original trial, study clinicians were assigned to their respective treatment conditions based on their allegiance to a specific approach for managing suicidal patients. Although this approach maximized fidelity to study treatment, it should be noted that CAMS therapists inadvertently had more than twice the years of practice experience since than ETAU therapists. This finding is consistent with research suggesting that less-experienced therapists may be more inclined to adhere to a specific treatment protocol or newly learned therapy whereas more seasoned providers may be more open to a variety of approaches [71-73]." 

4. Cost Assessment

In some areas the authors argue for using more realistic costs (e.g., onsite training); however, this is not consistent. An example is using the salary and fringe rates for pre-doctoral graduate students who would probably not be the clinicians if this was not a research study.

We acknowledge some inconsistency in the use of micro-costing techniques versus applying real-world assumptions. As noted as a limitation in the Discussion section, strict micro-costing procedures may exaggerate the costs of the CAMS condition owing to research activities performed for scientific rigor, above and beyond typical CAMS practice. Therefore, in some instances we applied assumptions that better reflect the costs of implementing CAMS in real-world settings. These assumptions are noted in the manuscript. A statement has been added to the Discussion section to emphasize this point: “We attempted to mitigate potential overestimation of costs related to research procedures through application of some real-world practice assumptions to enhance external validity.”

With regard to the pre-doctoral graduate students included in the cost assessment, the role of these individuals was limited to participating in consultation calls where feedback was provided related to treatment adherence; they did not serve as study clinicians. [As described in the Methods/Cost Assessment section, “Study therapists were clinical social workers assumed to be GS-12 paygrade based on key informant interviews.” To clarify this distinction, the section was revised to read “Annual base salaries and fringe rates for four pre-doctoral graduate student assistants who contributed to fidelity assessment were estimated based on typical grant-funding allowances reported by The Catholic University of America Office of Sponsored Programs.”]. We chose to micro-cost these activities based on the actual salary and fringe rates of pre-doctoral graduate students because (1) it represented a relatively small proportion of training and implementation costs (approximately 4 person hours total) and (2) we presume these cost rates are comparable to those of staff who might conduct administrative activities in real-world settings. 

What were primary hypotheses and primary study outcomes? Findings are presented for 6 months and 12 months, and for different cost outcomes and different cost variables. It is recommended that the primary outcome(s) be included in the Introduction, with a more specific statement of study hypotheses regarding the cost analysis variable(s) and outcome time point. Then, it would be helpful to organize the Results in keeping with this, presenting the primary outcomes first followed by any follow-up analyses.

As noted in an earlier response to the Reviewer, we have revised the Introduction section to specifically label the five study hypotheses, including the specific timepoints at which they were assessed. The Reviewer is correct that variables were analyzed at different timepoints; repeated measures, between-subjects variables were assessed at each follow-up timepoint (1-, 3-, 6-, and 12-month), whereas variables representing a within-subjects outcome were assessed only at 12-month follow-up. The latter was necessary because baseline (i.e., pre-intervention) healthcare expenditures were assessed on the THI-M over a “past year” timeframe, and therefore follow-up (post-intervention) healthcare utilization was only comparable after an equivalent 12-month period. These differences in assessment timeframes on the THI-M at baseline versus follow-ups are described in the Methods/Measures section, and we have specifically labeled the outcome as change in past-year expenditures. In addition, the following text was added to the Methods/Analyses section to provide clarification: “Linear regression was used to examine group differences in cost-benefit and net benefit at 12-month follow-up, as these outcomes reflect pre-intervention versus post-intervention change in past-year expenditures [54].” We are happy to include further clarification in the manuscript regarding outcome timepoints if deemed necessary.

We further made minor revisions to the Results section to be better organized in accordance with our primary outcomes.

5. Discussion and Conclusions. It is recommended that these focus on primary hypotheses and associated results and include as balanced as possible a presentation of findings.

The Discussion section has been reviewed to ensure focus on the results of primary hypotheses. Additionally, minor revisions to language have been made to present findings in a more neutral and balanced manner (e.g., expanding the implications and future directions of our work to suicide prevention more broadly).

6. PLOS authors have the option to publish the peer review history of their article (what does this mean?). If published, this will include your full peer review and any attached files.

Do you want your identity to be public for this peer review? For information about this choice, including consent withdrawal, please see our Privacy Policy.

Reviewer #1: No

Reviewer #2 Comments

1. If the authors have adequately addressed your comments raised in a previous round of review and you feel that this manuscript is now acceptable for publication, you may indicate that here to bypass the “Comments to the Author” section, enter your conflict of interest statement in the “Confidential to Editor” section, and submit your "Accept" recommendation.

Reviewer #2: (No Response)

2. Is the manuscript technically sound, and do the data support the conclusions?

Reviewer #2: Yes

3. Has the statistical analysis been performed appropriately and rigorously? 

Reviewer #2: I Don't Know

4. Have the authors made all data underlying the findings in their manuscript fully available?

Reviewer #2: Yes

5. Is the manuscript presented in an intelligible fashion and written in standard English?

Reviewer #2: Yes

6. Review Comments to the Author

Reviewer #2: Thank you for the opportunity to review the revised manuscript, “Costs, Benefits, and Cost-Benefit of Collaborative Assessment and Management of Suicidality versus Enhanced Treatment as Usual.” I was not among the initial reviewers from the first submission, and so did a ‘clean’ review of the new manuscript first, before examining the previous requests/responses to the initial review and revising my recommendations based on things that had already been addressed/discussed. Overall, the manuscript is well-written and more research is certainly needed regarding cost and cost-benefits of suicide-focused interventions, such as CAMS. I have outlined my specific recommendations and feedback below:

1) I was a bit confused when first reading the abstract, particularly the results reported on lines 34-38. They seem to indicate that CAMS had a significantly greater cost-benefit, but that this wasn’t significant when factoring in the intervention cost—wouldn’t intervention cost be factored into the cost-benefit? This part of the abstract may need to be reworded to clarify differences that are statistically significant. I also ran into this problem in the main manuscript where it was at times unclear whether a finding was suggesting a statistically significant difference between E-TAU and CAMS, or just trending in a particular direction.

We agree that these lines in the Abstract are a bit confusing. As originally worded, it is not clear how the cost-benefit metrics used (i.e., CBRs and net benefit) reflect different relationships between cost and benefit. To clarify, we have revised the Abstract to read, “CAMS also demonstrated significantly greater cost-benefit ratios (i.e., benefit per dollar spent on treatment) and net-benefit (i.e., total benefit less the cost of treatment) at 12-month follow-up.” Additionally, Reviewer #2 is correct that the term “significant” was misapplied on line 37. We have removed this statement and revised the Results and Discussion sections of the manuscript to clarify findings that were descriptive rather than statistically significant.

2) It was stated that the E-TAU group required a minimum of 4 sessions, and that CAMS consists of “approximately” 4-11 sessions. Was there a requirement of at least 4 sessions for the CAMS group? If not, this might present a conflict in the costs (given that ETAU needed 4 to be included). If the use of approximately meant that 4 and 11 represented the minimum and maximum range in the study, I would recommend making this explicit.

A minimum of 4 sessions was required in both treatments. The CAMS intervention requires a minimum of 4 sessions (initial plus three consecutive sessions without suicidality); thus, in the trial, ETAU was augmented with the same minimum for attention control. We have revised the Methods section of the manuscript to clarify that CAMS entailed a minimum of 4 sessions: “In the current study, the CAMS intervention consisted of approximately 4 to 11 weekly individual sessions (after the initial session, CAMS concludes after three consecutive sessions with resolved suicidality) following the suicide-specific CAMS framework [11].”

3) I was initially a bit surprised by the report that the cost of the CAMS intervention was not significantly different from E-TAU (given that there is a clear cost to CAMS, and not for E-TAU), and I’m a bit skeptical of the denominator used (160) for the number of patients treated with CAMS for each provider trained. It was described as being derived from suicidal ideation rates in the military population, the typical caseload size of a clinician, and the typical length of career for MHS behavioral health providers. Does this then assume that a clinician would be utilizing CAMS for every case of SI, with full fidelity? Many conditions (e.g., depression, PTSD) might have co-occurring SI where treating the SI may not be the primary treatment target. And with regard to the estimated career length, is this accounting for years already ‘served’ by the study therapists, or was this calculated as though training occurred for a new hire out of school? There are certainly differences in fidelity and adherence when in a consultation phase of an active RCT versus standard practice over a career, and I’d hesitate to extend the CAMS delivery with fidelity beyond a few years from the training. If there is existing evidence from CAMS or other EBPs regarding longevity of fidelity that supports the current estimates, please do cite and include. Otherwise, I would recommend revising down the current estimate of 160 patients being provided with fidelity-level CAMS, as it seems likely to be an overestimate that has significant implications for the rest of the cost equation.

The current analysis assumes that CAMS could be administered to nearly all patients presenting with suicidal ideation and the rate of appropriate patients was informed by both empirical evidence and key informant interviews with local providers. Treatment for co-occurring mental health conditions does not preclude concurrent treatment with CAMS, and in fact, CAMS is still indicated given empirical evidence suggesting that effective treatment of suicidal behaviors requires a suicide-specific focus (versus treatment focused on targeting a mental disorder with the reduction of suicidality as a secondary effect; see reviews by Wenzel, Brown, & Beck, 2008; Tarrier, Taylor, & Gooding, 2008). Thus, our analysis considered each suicidal patient as a potential candidate for CAMS regardless of the possibility that co-occurring conditions were the primary presenting concern. 

The analysis further assumes CAMS will be provided with an acceptable degree of fidelity. Reviewer #2 makes an excellent point that adherence may vary between an active RCT and standard practice. However, we anticipated that CAMS providers would remain generally adherent given (a) the lack of therapeutic drift in the primary RCT (see Jobes et al., 2017); (b) previous research from an online survey of practitioners trained in CAMS that indicated generally high levels of adherence to CAMS therapeutic approach and practice (Crowley, 2015); (c) the flexibility and adaptability of the CAMS framework to accommodate different suicidal patients and presentations; and (d) the relatively short Military Health System career span (estimated to be ~five years, conservatively). With regard to the estimated career length, our analyses assume that training would be provided at the beginning of the provider’s career such that it might be implemented for the full five years, which we believe to be a very conservative estimate of typical career length across both military and civilian providers.

Per Reviewer #2’s comment, we have added a statement regarding our assumption of maintained adherence to CAMS: “These estimates also assume a relatively high degree of adherence to CAMS across the provider’s MHS career, as supported by the absence of drift reported in the primary trial [24] and a community survey of mental health practitioners reporting generally high levels of adherence to the CAMS therapeutic approach and practice [53].”

4) I would recommend adding a quick note to the ‘cost’ section that the numbers also include 4 clinicians per training when arriving at the $12.44.

The Results/Cost section and Table 1 have been revised to specify that four clinicians were included in costing estimates for CAMS training and consultation activities.

5) Was there any data on whether these active duty soldiers were medically or otherwise discharged during this one-year period? If they were, would this have precluded them from receiving services that could have been captured by the cost analysis? If discharge rates differed between groups, and services would have been obtained outside of the DoD, this might look like cost savings, when in actuality, represents a more severe (and costly) outcome.

The Treatment History Interview (Military Version) captured all healthcare services within an assessment window regardless of where services were received; thus, the cost analysis reflects all services received, even if a participant separated from the military during follow-up. For reference, separation rates were comparable between conditions (53% separated from military services, 37.3% remained on active duty, and 9.3% status unknown in the ETAU condition versus (47.9% separated from military services, 43.8% remained on active duty, and 8.2% status unknown in the CAMS condition).

7. PLOS authors have the option to publish the peer review history of their article (what does this mean?). If published, this will include your full peer review and any attached files.

Do you want your identity to be public for this peer review? For information about this choice, including consent withdrawal, please see our Privacy Policy.

Reviewer #2: No

REVIEWS RECEIVED 6/11/2021

Journal Requirements:

We have reviewed our reference list and believe it to be complete and correct. Two citations in the Introduction section have been replaced with more appropriate citations in the revised manuscript; both instances are marked with a comment in the ‘track changes’ version of the manuscript. Additionally, several new citations have been added to support revisions requested by reviewers; these instances are marked in red font.

We have reviewed the style requirements and believe our submission to be consistent with the guidance provided.

"I have read the journal's policy and the authors of this manuscript have the following competing interests: David Jobes has conflicts to disclose related to grant funding from the National Institute of Mental Health; he receives book royalties from the American Psychological Association Press and Guilford Press; and he is the founder of CAMS-care, LLC (a professional training and consultation company). Phoebe McCutchan, Brian Yates, Amanda Kerbrat, and Katherine Comtois have declared that no competing interests exist."

The authors confirm that competing interests do not alter our adherence to all PLOS ONE policies on sharing data and materials. We have included the statement, “This does not alter our adherence to PLOS ONE policies on sharing data and materials” in our revised cover letter. 

There are ethical and/or legal restrictions that prohibit sharing of a de-identified dataset. We have added the following data availability statement to our revised cover letter (and have also provided this statement where prompted in the electronic submission portal): “Public sharing of data used in this study is prohibited under the protocol approved by the U.S. Army Medical Research and Development Command Office of Research Protections as well as Institutional Review Boards at Dwight D. Eisenhower Army Medical Center, The Catholic University of America, and the University of Washington. Additional restrictions apply to the availability of these data in order to protect participants’ privacy. Requests for access from interested researchers may nevertheless be considered, subject to the terms and conditions of the request and in compliance with the applicable regulations. Requests may be directed to the Office of Sponsored Programs and Research Services, The Catholic University of America, 213 McMahon Hall, 620 Michigan Ave., NE, Washington, DC 20064; Phone: 202-319-5218; Fax: 202-319-4495; Email: CUA-OSP@cua.edu.”

REVIEWERS' COMMENTS:

Reviewer's Responses to Questions

Comments to the Author

1. Is the manuscript technically sound, and do the data support the conclusions?

Reviewer #1: Yes

Reviewer #2: Yes

2. Has the statistical analysis been performed appropriately and rigorously?

Reviewer #1: I Don't Know

Reviewer #2: Yes

3. Have the authors made all data underlying the findings in their manuscript fully available?

Reviewer #1: Yes

Reviewer #2: Yes

4. Is the manuscript presented in an intelligible fashion and written in standard English?

Reviewer #1: Yes

Reviewer #2: Yes

5. Review Comments to the Author

Reviewer #1: 

Article well written and well presented. I have only a few remarks to propose. This article is focused on a medico-economic analysis of a prevention program (CAMS) on suicidal behaviors in suicidal soldiers of the US army. This intervention is delivered by social workers, and is compared to an Enhanced version of Treatment As Usual ETAU). I would be interested if some items could be further discussed:

1) Social workers are not randomized and in CAMS program are far more experienced. It is only discussed in the limitations part saying it has no influence. Some articles emphasize that experience might be important. (For instance: Suicide intervention skills and related factors in community and health professionals Suicide Life Threat Behav . 2010 Apr;40(2):115-24. doi: 10.1521/suli.2010.40.2.115.)

We appreciate the reviewer’s comment and agree that this limitation could be presented in a more balanced manner given research suggesting that suicide-specific practice experience (although not necessarily general practice experience) is associated with greater suicide prevention skills. We have revised this paragraph of the Limitations section to read, “Still, some studies indicate that greater suicide-specific practice experience is associated with enhanced suicide intervention competence and skills [77, 78], and thus future CAMS investigations would be strengthened by ensuring comparable years of practice experience among providers to remove this potential confounding.”

2) CAMS protocol stops if 3 consultations in a row are without suicidal ideas, as ETAU does not. This might have direct economic implications.

The randomized controlled trial which provided data for the current study was designed to examine real-world effectiveness. The CAMS protocol in the current study was delivered in accordance with the standardized treatment manual typically used in CAMS implementation (i.e., the criteria for treatment termination were not specific to this study). ETAU clinicians concluded study treatment when the suicidal risk for which they were referred resolved, as defined by their and the clinic’s standard practices. Once study treatment concluded, they could continue to see or refer the participant for other treatment issues or discharge the participant from treatment. This ‘real world’ comparison control condition was intentionally chosen to maximize the external validity and generalizability of the study, to include the estimation of treatment costs for comparison. The purpose of the current study was to examine the economic implications of providing each of these treatment interventions under real world conditions. 

Interestingly, as described in the manuscript, there were no significant differences in the number of study treatment sessions between CAMS and ETAU, indicating that treatment length was comparable between groups and likely not impacted by discrepant treatment termination protocols.

3) It seems that Crisis Services use is different in the 2 programs and seems more important in CAMS program in the 1st month, and less at 6 months and 12 months. This seems very interesting and should be further discussed.

We have revised the Discussion section to read, “With regard to benefits at the group level, CAMS was associated with significantly reduced total cumulative healthcare expenditure compared to ETAU at 6-month follow-up, likely driven by significantly lower crisis services expenditures. However, ETAU eventually matched CAMS in total cumulative healthcare expenditures at 12-month follow-up timepoints, although crisis services expenditures remained significantly lower for CAMS. The data suggest that participants in the CAMS condition may use more crisis services than ETAU in the first month of treatment but use significantly fewer crisis services by 6- and 12-month follow-ups. The early uptick in crisis service utilization may be explained by CAMS's suicide-specific focus which promotes accessing such services in an acute crisis, whereas the decreased utilization by six months likely reflects findings from the primary trial that CAMS participants had a lower probability of having suicidal ideation at 3-month follow-up [24].”

Reviewer #2: 

Thank you for the opportunity to review this interesting paper. Similarly to reviewer 2, I was not among the initial reviewers and carried out a first blind review of the manuscript before considering the authors’ responses to the initial review and revising my recommendations based on things that had already been addressed or discussed (which contributes to the reduced number of comments I have). Overall, I believe that this article is of research and policy interest, rigorous, well-written and clear. I, however, still have a couple of comments to improve the latest version of the manuscript but want to stress that in my view the comments of previous reviewers have been appropriately addressed overall.

1/ I believe that, for non-US readers, some parts could be made clearer or more specific. In particular, behavioral health is a concept which is very specific to the US and which would be worth defining once. 

We have revised the manuscript to define ‘behavioral health’ the first time it appears in the text: “Participants were randomly assigned to either CAMS or ETAU matched on histories of suicide attempts, medication class, severity of physical injury or disability, and current enrollment in outpatient behavioral health treatment (i.e., psychiatric, clinical psychology, or social work services).”

The involvement of social workers in mental care is also very specific to your national context and could require some additional information to justify why you used social workers wages in your economic evaluation. 

Social worker wages were used in this economic evaluation to reflect the actual wages received by the therapists employed in the study, in line with our micro-costing approach. The credentials of the providers are described under the Methods/Participants section and the Cost Assessment/Training and implementation activity costs/Estimating resource inputs and unit costs section.

We acknowledge that our cost estimates may not generalize to all healthcare contexts and have noted this as a limitation in the Discussion section. In response to the reviewer’s comment, we have added an additional sentence emphasizing that costs may be influenced by therapists’ credentials: “Still, costs of real-world CAMS delivery may be substantially lower in some settings owing to the availability of less expensive training formats (e.g., online video courses; www.CAMS-care.com) and delivery formats (e.g., group therapy) [64], as well as its suitability for use by diverse types of providers (e.g., paraprofessionals) [13]. Conversely, costs may be greater in settings where outpatient mental health services are typically delivered by doctoral-level therapists. Additional cost-inclusive evaluations of CAMS across these various implementation approaches are needed to inform wider dissemination of the framework.”

In the introduction, it would be interesting to provide more specific figures on the unprecedented rate of increase in suicides among the military workforce in the US over time since such figures appear to be available. 

We have added the rate of increase to the manuscript, in addition to updating the suicide mortality rate to reflect estimates published since the initial submission of this manuscript: “Active duty service members have demonstrated continuously rising suicide rates across all branches of service. In 2019, the suicide mortality rate across all active duty services combined was 25.9 per 100,000, representing a per-year rate ratio of 1.04 from calendar years 2011 through 2019 [3].” 

Also briefly mention what is TRICARE and CHAMPUS. 

We have revised the manuscript to read, “Each visit was assigned an appropriate Current Procedural Terminology (CPT) [54] and costed using corresponding 2018 TRICARE/CHAMPUS Maximum Allowable Charges based on facility, non-physician provider, and appropriate locality rates. TRICARE (formerly known as CHAMPUS) is a DoD health insurance program, and its Maximum Allowable Charges rates reflect costs directly related to provision of the service (i.e., provider time used in the visit), practice expenses such as facilities and administrative staff, and malpractice insurance.”

Minor additional precisions needed: 

“Active component”: clarify for readers what this refers to;

We have revised the first mention of this term in the manuscript to read, “In 2019, the suicide mortality rate for the Active Component (i.e., full-time service members) across all services was 25.9…”

 “condition”: maybe replace everywhere by “treatment condition” as you do once, that way it is less confusing for non-native English speakers (as conditions can refer to diseases);

The manuscript has been revised to replace “condition” with “treatment condition” except in instances where the treatment condition is specified (i.e., CAMS condition or ETAU condition).

 “modeling on multiply imputed datasets” (abstract): multiple?, 

We believe “multiply” to be the appropriate term; specifically, the analytic procedure of ‘multiple imputation’ produces ‘multiply imputed’ data.

“member of the Warriors in Transition unit”: explain what it is very briefly?

We have added the following description to the manuscript, “Warrior Transition Unit (a unit providing support to soldiers being treated for chronic and/or severe injuries who cannot yet return to work)”

2/ I think the discussion or conclusion could stress more the policy implications of the research for the military health system (make it more explicit maybe).

We have added to the Conclusions section, “These findings may inform more robust cost-inclusive analyses in future trials and ultimately facilitate decision-making among the policymakers, healthcare administrators, and clinicians tasked with selecting mental health programs that will provide the biggest ‘bang for the buck’ amidst constrained resources.”

3/ I am surprised that none of the ETAU clinicians, when provided with potential supervision upon request, asked for it. Maybe discuss potential hypotheses about why this happened (while it seems to be less the case with CAMS while it is carried out by providers with more years of experience).

We have added the following hypothesis to the Methods/Cost Assessment/Training and implementation activity costs section: “Further, although ETAU therapists were offered additional supervision and consultation as needed, key informant interviews indicated that they elected not to participate in these activities during the course of the study. This may reflect stronger allegiance to a specific treatment approach for managing suicidality.”

4/ At some point you mention the time required for transit between the patient’s place of residence and the place of care and justify that you did not include it in your economic evaluation because you were not able to access confidential information regarding the place of residence of patients. I wonder if anyway this cost should be included in an evaluation adopting the military health services perspective? Would not those costs lie on the patients anyway or be more related to loss of productivity for the military?

The reviewer is correct that participant time/travel costs are not pertinent to an economic analysis from a healthcare payer perspective (such as the current study). We have removed this text from the manuscript to avoid confusion.

5/ I read your answer to a previous comment on the pre-doctoral graduate students included in the cost assessment. I am still not fully convinced they should be included in an evaluation which aims to accurately fit real-world settings. “We presume these cost rates are comparable to those of staff who might conduct administrative activities in real-world settings.”: This should be more justified to be convincing.

We appreciate the concern that inclusion of the pre-doctoral graduate student activities may not entirely reflect real-world settings. However, we felt the need to include these activities as part of our micro-costing procedures because the attention to fidelity may have had an impact on the overall effectiveness (and thus economic outcomes) associated with CAMS treatment. We also maintain that such fidelity monitoring activities may be performed as part of real-world CAMS implementation, for example as part of training new hires. In such instances, it is reasonable to assume that such activities may be performed by supervisory staff using the CAMS Rating Scale. Additionally, while the individual cost rates used in the study may not entirely represent real-world settings, we believe that the aggregate of Dr. Jobes’ estimated salary rate (most expensive) and graduate students’ estimated salary rate (less expensive) likely captures a cost rate comparable to a typical supervisor’s time.

6/ Some reorganization might be necessary: “In the current secondary analysis, treatment costs […] were determined using micro-costing” and subsequent paragraph of the introduction: this would be better suited in the method section; 

We believe that including a brief methodological description in the Introduction allows for definition of key concepts which may be unfamiliar to readers unacquainted with economic assessment and provide context for the hypotheses presented. 

“Multiple imputation using chained equations with predictive mean matching was used to minimize uncertainty…”: this would be better suited in the method section of the paper as well.

Our description of multiple imputation procedures has been moved to Materials and methods/Analyses section.

7/ For the ethical approvals that you mention in the first paragraph of the method section, it would better (more transparent and traceable) to also provide the identification number of the approval received.

We have added the ID number of the current archival study reviewed by the American University IRB.

8/ Measures section: “were validated against available administrative data sources (e.g., electronic health records”: it would be better to be exhaustive and more specific here; 

We have replaced “e.g.,” in this instance with “i.e.,”.

and the comma after e.g. (or sometimes i.e. in the text) seems misplaced.

We respectfully maintain that the commas in “e.g.,” and “i.e.,” are consistent with the PLOS ONE style but are happy to revise upon the editorial board’s request.

 Similarly, in the ‘Training and implementation activity costs’ paragraph: “Activities performed were identified by intervention manuals, administrative data systems, and qualitative interviews as feasible and appropriate”: it would be better to be more precise here.

We have revised this description to read, “Activities performed were identified by study protocols, intervention manuals, administrative data systems maintained by research personnel, and qualitative interviews with research personnel and study therapists as feasible and appropriate.”

9/ “There were no statistically significant differences between the CAMS and ETAU conditions with regard to sociodemographic or baseline clinical characteristics”: maybe specify that you are talking about patients’ characteristics.

We have revised this sentence to read, “There were no statistically significant differences between the CAMS and ETAU conditions with regard to patient participants’ sociodemographic or baseline clinical characteristics.”

10/ I must admit I am a bit concerned about the involvement of the founder of a company called CAMS-care in a research which strongly supports the overall value of CAMS.

Dr. Jobes, the pioneer of the CAMS intervention, has plainly and clearly disclosed his competing interests in accordance with PLOS ONE policy. We also refer the Reviewer to a recent meta-analysis finding little to no publication or allegiance bias in the broader evidence base supporting CAMS as a treatment for suicidal ideation (Swift JK, Trusty WT, Penix EA. The effectiveness of the Collaborative Assessment and Management of Suicidality (CAMS) compared to alternative treatment conditions: A meta-analysis. Suicide Life Threat Behav. 2021. Epub 2021/05/18).

6. PLOS authors have the option to publish the peer review history of their article (what does this mean?). If published, this will include your full peer review and any attached files.

Do you want your identity to be public for this peer review? For information about this choice, including consent withdrawal, please see our Privacy Policy.

Reviewer #1: No

Reviewer #2: No

---

## [Editor Report · Decision Letter 1]

30 Dec 2021

Costs, Benefits, and Cost-Benefit of Collaborative Assessment and Management of Suicidality versus Enhanced Treatment as Usual

PONE-D-21-12016R1

Dear Dr. McCutchan,

We’re pleased to inform you that your manuscript has been judged scientifically suitable for publication and will be formally accepted for publication once it meets all outstanding technical requirements.

Kind regards,

Isabelle Durand-Zaleski

Academic Editor

PLOS ONE
---

## [Editor Report · Acceptance letter]

25 Jan 2022

PONE-D-21-12016R1 

Costs, Benefits, and Cost-Benefit of Collaborative Assessment and Management of Suicidality versus Enhanced Treatment as Usual 

Dear Dr. McCutchan:

I'm pleased to inform you that your manuscript has been deemed suitable for publication in PLOS ONE. Congratulations! Your manuscript is now with our production department. 

Kind regards, 

on behalf of

Dr. Isabelle Durand-Zaleski 

Academic Editor

PLOS ONE